# Identification of *ZmBK2* Gene Variation Involved in Regulating Maize Brittleness

**DOI:** 10.3390/genes14061126

**Published:** 2023-05-23

**Authors:** Wei Xu, Yan Zhao, Qingzhi Liu, Yuqiang Diao, Qingkang Wang, Jiamin Yu, Enjun Jiang, Yongzhong Zhang, Baoshen Liu

**Affiliations:** 1Agronomy/State Key Laboratory of Crop Biology, Shandong Agricultural University, Tai’an 271000, China; xwnxn2016@163.com (W.X.);; 2Qingdao Academy of Agricultural Sciences, Qingdao 266100, China; 3Taian Denghai Wuyue Taishan Seed Industry Co., Ltd., Tai’an 271000, China

**Keywords:** lodging resistance, brittle mutants, *ZmBK2*, COBRA-like proteins, cell walls

## Abstract

Maize stalk strength is a crucial agronomic trait that affects lodging resistance. We used map-based cloning and allelic tests to identify a maize mutant associated with decreased stalk strength and confirmed that the mutated gene, *ZmBK2*, is a homolog of *Arabidopsis AtCOBL4*, which encodes a COBRA-like glycosylphosphatidylinositol (GPI)-anchored protein. The *bk2* mutant exhibited lower cellulose content and whole-plant brittleness. Microscopic observations showed that sclerenchymatous cells were reduced in number and had thinner cell walls, suggesting that *ZmBK2* affects the development of cell walls. Transcriptome sequencing of differentially expressed genes in the leaves and stalks revealed substantial changes in the genes associated with cell wall development. We constructed a cell wall regulatory network using these differentially expressed genes, which revealed that abnormal cellulose synthesis may be a reason for brittleness. These results reinforce our understanding of cell wall development and provide a foundation for studying the mechanisms underlying maize lodging resistance.

## 1. Introduction

Lodging threatens maize harvests and reduces the total annual yield by 5–20% worldwide, increasing the risk of food insecurity [1]. Lodging resistance is closely related to plant brittleness, which involves complex physicochemical processes associated with cell wall development, specifically cellulose and lignin biosynthesis [2]. Research on brittle mutants is important for understanding the regulatory mechanisms of plant mechanical strength and has major implications for crop lodging resistance breeding [3].

The plant cell wall acts as a rigid and dynamic network that primarily consists of polysaccharides and supports plant cell morphogenesis. Cell walls are mainly composed of cellulose, hemicellulose, pectin, and lignin, with cellulose being the main component in most plants [4]. Cellulose consists of unbranched β-(1,4)-linked glucan chains and provides support for cell walls. According to their structural and compositional characteristics, cell walls are classified into primary and secondary cell walls [4]. Primary cell walls form during cytokinesis and modify and expand with cell growth, providing strength, resilience, and plasticity to the cell [5]. Secondary cell walls develop between the primary cell wall and protoplast after the cell stops growing; they have a cellulose content of 40–90% and also contain other principal components such as lignin. Primary and secondary cell walls contribute greatly to the mechanical strength of a plant [6]. Rice mutants *Tos17*, *bc7*, and *bc11* have defects in cellulose synthase catalytic subunit (*CESA*) genes, causing reduced cellulose content in secondary cell walls and decreased mechanical strength [7]. *Arabidopsis* irregular xylem mutants *irx1* and *irx3* exhibit a disordered secondary cell wall structure [8], suggesting that cellulose deposition plays a key role in the integrity and assembly of cell wall polymers.

Studies on brittle mutants have identified some genes involved in cellulose biosynthesis [9,10]. In higher plants, cellulose synthesis is generally catalyzed by cellulose synthase complexes that form a rosette structure and synthesize at the plasma membrane (PM) using uridine diphosphate (UDP)-glucose as substrates [11]. The plant *CESA* gene was first isolated from cotton fiber based on its high similarity to bacterial *CESA* sequences. Most *CESAs* are highly conserved and co-expressed to regulate primary or secondary cell wall synthesis with multispan membrane domains [12,13]. Twelve *CESAs* have been cloned in *Arabidopsis*; of these, seven are required for primary cell wall synthesis, five are involved in secondary cell wall metabolism, and the functions of the remaining two are unknown [8,10]. Rice contains 11 *CESAs* that play key roles in secondary cell wall synthesis, with most of them being homologs of *AtCESAs* [14]. Of the twelve *CESAs* that were separated from maize, *ZmCESA1* to *ZmCESA9* are involved in primary cell wall formation, and *ZmCESA10* to *ZmCESA12* interact to form enzyme complexes that participate in secondary cell wall synthesis [15].

Additional proteins, such as COBRA, are necessary for the synthesis and orderly deposition of cellulose. First identified in *Arabidopsis*, the COBRA mutant had decreased cellulose content and exhibited plant brittleness [16]. The COBRA family genes encode glycosylphosphatidylinositol (GPI)-anchored plant-specific proteins with four conserved structural domains: a carbohydrate-binding motif, N-terminal protein targeting sites, a CCVS (Cys-rich) motif, and a hydrophobic C-terminus with an attachment site [17]. When the GPI-anchor modification site (ω-site) is cleaved, this structure combines with the C-terminus of the COBRA protein via an amide bond [18]. Research on *Arabidopsis cob-1*, *cob-2*, and *cob-3* revealed that the COBRA protein affected the direction of cell expansion and decreased crystalline cellulose content [17]. The COBRA proteins play extremely important roles in anisotropic cell expansion and influence the morphogenesis of the whole plant by directing cellulose microfibril formation [19]. Rice *bc1* reduces glucose and xylose and alters cellulose and lignin contents, resulting in defective development of secondary cell walls, which suggests that it may act as an auxiliary protein in regulating secondary cell wall biosynthesis [20]. Maize *ZmBK1*–*ZmBK6* were identified to be associated with defective cell wall development, and *ZmBK2* was first identified in maize as a member of the COBRA family [21,22,23]. The *ZmBK2* gene greatly reduces tissue mechanical strength, along with a substantial reduction in cellulose content and impaired deposition of secondary cell wall components [22,24].

A maize *brittle stalk mutant* (*bk2*) was identified and exhibited reduced mechanical strength, decreased cellulose content, and abnormal cell wall morphology in the whole plant. According to homology analyses, *ZmBK2* encodes a GPI-anchored plant-specific protein that participates in cellular metabolic processes. We identified a series of genes concerned with cell wall development using transcriptome analysis and constructed a putative regulatory network for cell wall development. This study illuminates the role of *ZmBK2* in cell wall biosynthesis and provides new insights into the mechanisms of maize brittleness.

## 2. Materials and Methods

### 2.1. Plant Materials

In this study, we analyzed three brittle mutants (*mu-J1862*, *mu-J184*, and *916C*) and their wild-type counterparts (J1862, J184, and W22). The spontaneous mutant *mu-J1862* was crossed with B73 to obtain F_1_ and backcrossed with *mu-J1862* to obtain BC_1_F_1_. Thereafter, BC_1_F_1_ was self-crossed three times to obtain a homozygous non-brittle plant, J1862. *mu-J184* was crossed with B73 to generate F_1_ and F_2_ for further research on homozygous brittle and non-brittle plants. To generate F_1_ and F_2_, *mu-J1862* was crossed with B73 and Huangzaosi, with B73 and Huangzaosi being chosen as the two inbred lines for the mapping population due to their significant genetic differences from the mutant. Brittle plants from the F_2_ population were used for genetic mapping. Maize stock *916C* containing the reference allele of *bk2* (*bk2*-*ref*) was obtained from the Maize Genetics COOP Stock Center (https://www.maizegdb.org/data_center/stock, accessed on 21 May 2021) and backcrossed three times with W22, with W22 serving as the background material. All materials were grown in Shandong and Hainan Provinces in China.

### 2.2. Map-Based Cloning of the Brittle Gene

We used 3107 plants with a brittle phenotype from the F_2_ population for fine mapping by the BSA method. Simple sequence repeat (SSR) primers were obtained from the Gramene database (http://archive.gramene.org/markers/microsat/, accessed on 10 February 2022). Sequence polymorphisms were screened between *mu-J1862* and B73 for the development of SSR and indel markers. Primer pairs were automatically designed in Primer3 (https://bioinfo.ut.ee/primer3-0.4.0/, accessed on 21 May 2022) and dCAPS Finder 2.0 (http://helix.wustl.edu/dcaps/dcaps.html, accessed on 15 July 2022). The Maize Genome Annotation Project (https://www.maizegdb.org/, accessed on 7 October 2022) was used for annotating putative genes.

### 2.3. Allelic Tests of bk2

Two allelic tests were performed by crossing *mu-J1862* with *mu-J184* and *mu-J1862* with *916C*.

### 2.4. Stalk Stiffness Test and Measurement of Stalk and Leaf Composition

Stalk stiffness was measured with a digital force gauge that recorded the maximum tensile force by pulling the specimen until it snapped.

To determine cellulose and lignin compositions, HPLC and UV spectrophotometry were performed based on the National Renewable Energy Laboratory (NREL/TP-510-42618) procedures [25]. Cellulose content was measured by HPLC based on degraded glucose and xylose sugar units and the amount of acid-soluble lignin by a UV spectrophotometer. Hemicellulose was measured according to Jin et al. [26]. Maize stalks and leaves were collected at the 12-leaf stage.

### 2.5. Subcellular Localization

The full-length CDS of *ZmBK2* was amplified from B73 and cloned into the transient expression vector Pzp211-GFP-*ZmBK2,* driven by a constitutive CaMV 35S promoter. Clones were introduced into tobacco leaves to generate fusion proteins. Green fluorescent signals were detected using a TCS SP5 II laser-scanning confocal microscope (Leica, Wetzlar, Germany).

### 2.6. Phylogenetic Analysis and Protein Modeling

Homologous *ZmBK2* amino acid sequences were searched through the NCBI database (https://blast.ncbi.nlm.nih.gov/Blast.cgi, accessed on 14 November 2022). Clustal Omega (http://www.clustal.org/, accessed on 14 November 2022) was used to perform sequence alignments. We employed MEGAX (https://www.megasoftware.net/, accessed on 14 November 2022) with default settings to reconstruct a phylogenetic tree, using 1000 bootstrap iterations to estimate the tree’s dependability. SWISS-MODEL (https://swissmodel.expasy.org/, accessed on 14 November 2022) was used to predict the protein structure of *ZmBK2*.

### 2.7. Transcriptome Analysis

Total RNA from the V5 stalks and leaves of *mu-J1862* and wild-type plants was extracted using the TRIzol method. Three biological replicates were performed, each consisting of mixed stalks or leaves from 10 plants. The cDNA library was constructed from 12 samples and sequenced on an Illumina HiSeq 4000 platform at Novogene Bioinformatics Technology (Beijing, China), The electrophoresis patterns of RNA in the 12 samples are presented in Appendix A, while the RIN values of the 12 RNA samples are shown in Appendix A. Clean reads were mapped to the maize reference genome (B73 Ref-Gen v5.0.54) in HISAT2 [27]. Differentially expressed genes (DEGs) were screened using DEseq2 [28], based on the criteria of fold-change ≥ 2 and FDR < 0.01. We used the GOseq package in R to assess the enrichment of Gene Ontology (GO) terms among DEGs [29]. A pathway-based analysis was performed using KEGG to further investigate genes’ biological functions and interactions. RNA-seq data were deposited in the NCBI Sequence Read Archive (SRA) under accession number SRP12182200 (BioProject ID: PRJNA896366).

### 2.8. RNA Extraction and RT-qPCR

Total RNA was extracted from tissue samples using 1 mL of TRIzol reagent (Thermo Fisher Scientific, Waltham, MA, USA), according to the manufacturer’s guidelines. After isopropanol precipitation, RNA was resuspended in 50 µL RNase-free water and treated with RNase-free DNase I. First-strand cDNA was reverse-transcribed from RNA (1 μg/20 μL) using the HiScript II 1st Strand cDNA Synthesis Kit (Vazyme, Nanjing, China). SYBR Green (Bio-Rad, Hercules, CA, USA) was added to PCR reactions, and three independent RNA samples were analyzed by RT-qPCR using the Bio-Rad CFX96 Touch Real-Time PCR detection system with *ZmActin1* as an internal control. The primers for RT-qPCR are listed in Appendix A, and relative transcript levels were calculated using the 2^−ΔΔCt^ method [30]. Melting Curve Analysis of the primers used are shown in Appendix A. To examine the expression pattern of *ZmBK2*, three-leaf stage leaf, three-leaf stage root, three-leaf stage stalk, thirteen-leaf stalk, above-ear leaf, lower-ear leaf, functional leaf, anther, filament, female spikelet, tassel, 18-day embryo, and 18-day endosperm were analyzed.

### 2.9. Putative Cell Wall Regulatory Network

Results from GO enrichment analyses were used to predict the *ZmBK2* regulatory network with STRING V11 (https://cn.string-db.org/, accessed on 14 January 2022) [31].

### 2.10. Phylogenetic Analysis of ZmBK2

To determine the evolutionary relationships of *ZmBK2* based on the protein sequences, a phylogenetic tree was generated using the Neighbour-Joining (NJ) algorithm.

## 3. Results

### 3.1. Brittle Mutant Decreased Maize Tissue Strength

Spontaneous maize mutants *mu-J1862* and *mu-J184* exhibited whole-plant brittleness, including the leaves, stalks, male spikes, ears, and roots. Field-grown mutants were susceptible to breakage even under mildly windy conditions. After V5, brittle mutant veins and leaves snapped and broke away cleanly, with a crisp sound and a clear cut after bending. As more stress was applied, the stalk lost its toughness and snapped as the internodes continued to bend. The wild-type stalk remained in a bent position but did not snap under constant stress (Figure 1A,B). After 24 days of pollination, we measured the residue rates of wild-type and mutant kernels, yielding 14.4% and 12.6%, respectively (Appendix A).

### 3.2. Genetic Analysis and Map-Based Cloning of the Brittle Gene

Two F_2_ populations derived from *mu-J1862* and wild-types (B73, Huang Zaosi) were used for genetic analyses. The segregation ratio of wild-type to mutant was approximately 3:1 (Table 1), indicating that a single recessive nuclear gene controlled the brittle phenotype. Preliminary linkage analysis suggested that the brittle gene was located between SSR markers *Xwd2-28* and *Xwd4-37* on the long arm of chromosome 9 (Figure 1C-a). After developing 10 SSR markers located in this interval, the brittle gene was further delimited to 31.1 Mb bracketed by *p-umc1570* and *p-umc2341* (Figure 1C-a). Fourteen SSR markers listed in Appendix A were designed between *p-umc1570* and *p-umc2341* for fine mapping. Using 3607 brittle plants from the two F_2_ populations of 11,850 plants, the brittle gene was finally narrowed down to a 147-kb region between markers *Xwd6-21* and *Xwd7-6* (Figure 1C-b). Four putative genes in the 147-kb region were annotated (Figure 1C-c). *Zm00001eb393060* encodes a PLC-like phosphodiesterase, *Zm00001eb393070* is the *ZmBK2* that has been reported to be associated with the brittleness of maize plants, and the function of *Zm00001eb393080* and *Zm00001eb393090* is unknown.

Previous studies on *bk2* showed that the abnormal cellulose deposits in the secondary cell wall dramatically reduced tissue mechanical strength [22,24]. Our results indicate that *ZmBK2* encodes a COBRA-like protein and contains two exons and one intron (Figure 1D). The primers were designed based on *ZmBK2* to amplify 2.31-kb fragments from the two mutants, with a 51-bp insertion at 1357 bp in *mu-J1862* and a 34-bp deletion at 1727 bp in *mu-J184* (Figure 1E). Both insertions and deletions altered the amino acid sequence and protein structure of BK2. Primers *BK2-J1862F/R* and *BK2-J184F/R* were then designed based on the deletion and insertion sequences (also presented in Appendix A). The 51-bp insertion and 34-bp deletion were not detected in 12 and 10 different inbred lines, respectively, indicating that the mutation sites were unique and could be used as markers to detect the brittle phenotype (Appendix A). The *BK2-J1862F/R* marker was verified in the backcross population (*mu-J1862*/B73//*mu-J1862*). Ten mutant plants exhibited a 252-bp band consistent with *mu-J1862*, while the 10 wild-type plants exhibited two bands consistent with B73 and *mu-J1862* (Appendix A).

### 3.3. Allelic Tests

To verify whether *Zm00001eb393070* was responsible for brittleness, *916C* with a transposon in the first exon was used (Figure 2A). The two F_1_ hybrids of *mu-J1862* × *mu-J184* and *mu-J1862* × *916C* exhibited brittleness (Figure 2B,C). Based on these results, *mu-J1862*, *mu-J184*, and *916C* were identified as brittle alleles.

### 3.4. Stalk Stiffness Test and Composition Measurements of Stalks and Leaves

Mutants (*mu-J1862*, *mu-J184*) and wild-types (J1862, J184) did not differ significantly in stalk thickness or 100-grain weight (Appendix A). Stiffness testing of the third above-ground internode in mutants was half of that of the wild-types (Figure 2D). The snapping tension of mutant and wild-type stalks was 18–32 N and 45–58 N, respectively.

We assayed stalks and leaves compositions of *mu-J1862*, *916C*, and *mu-J1862*/*916C*. Mutant plants had significantly lower cellulose and hemicellulose content in stalks and leaves than wild-type plants. Lignin content decreased in leaves and slightly increased in stalks (Figure 2E–G).

### 3.5. Brittle Mutant Exhibits Abnormalities in Sclerenchyma Cells

Mutant plants exhibited brittle phenotypes and reduced mechanical strength, suggesting that the structure of their secondary cell walls may be abnormal. In *mu-J1862*, *mu-J184*, and *mu-J1862*/*916C* leaves, sclerenchyma had thinner cell walls and were fewer in number, being present at 63.9%, 65.4%, and 57.9% of their corresponding wild-type number, respectively (Figure 3A–C). This abnormal development caused insufficient leaf support and negatively affected the mechanical strength of the leaves. The lacuna of mutant stalks was more compact, and safranin O staining in the vascular bundle was darker, indicating that lignin content had increased (Figure 3D).

### 3.6. Constitutive Expression, Subcellular Localization and Phylogeny of ZmBK2

The results of RT-qPCR indicated that *ZmBK2* was highly expressed in three-leaf stage stalks (Figure 4A). Confocal microscopy showed that the *ZmBK2* protein was predominantly localized to the cytoplasm of the cell (Figure 4B).

The phylogenetic analysis via NJ algorithm showed that the most homologous gene of *ZmBK2* was *AtCOBL4*, which belongs to the COBRA family (Figure 4C).

### 3.7. Transcriptome Alterations in mu-J1862 Stalks and Leaves

To examine the genome-wide effects of *ZmBK2*, we used RNA-seq to compare the transcriptome profiles of *mu-J1862* and J1862. A total of 78.62 Gb original data was obtained from cDNA libraries. Approximately 79% of clean reads were unique and mapped to the maize B73 genome (Appendix A).

In mutant stalks and leaves, 1694 and 1431 genes were up-regulated, and 2044 and 1729 genes were down-regulated (Figure 5A,B). Down-regulated genes in mutant stalks were mainly enriched in secondary metabolic processes, phenol-containing compound metabolic processes, metabolic processes, and phenylpropanoid biosynthesis (Appendix A). Down-regulated genes in mutant leaves were enriched in secondary metabolite biosynthesis and cyanoamino acid metabolism (Appendix A). Up-regulated genes in stalks were mainly enriched in riboflavin metabolism, phenylpropanoid metabolism, and secondary metabolic processes (Appendix A). Up-regulated genes in leaves were enriched in secondary metabolite biosynthesis, metabolic pathways, and phenylpropanoid metabolism (Appendix A). Up-regulated and down-regulated genes in stalks and leaves were co-enriched in secondary regulatory and phenylpropanoid metabolic pathways. Products of secondary regulatory pathways were essential for cell wall synthesis, and phenylalanine was the starting point for cellulose synthesis in the phenylpropanoid metabolic pathway. These results suggest that *ZmBK2* may influence the expression of cell wall-related genes and that *ZmBK2* may work together to regulate the development of plant cell walls.

We analyzed co-up-regulated and co-down-regulated genes in both stalks and leaves. Compared with wild-types, 376 genes were co-down-regulated and 276 genes were co-up-regulated in mutant leaves and stalks (Figure 5C). Co-down-regulated genes were mainly enriched in molybdopterin cofactor biosynthetic processes (GO:0019720), polysaccharide metabolic processes (GO:0005976), cell wall biogenesis (GO:0042546), and cell wall macromolecule metabolic processes (GO:0044036) (Figure 6A), while co-up-regulated genes were mainly enriched in reproductive shoot system development (GO:0090567), developmental processes involved in reproduction (GO:0003006), cell wall organization or biogenesis (GO:0071554), cellular response to ammonium ions (GO:0071242), and cellulose biosynthetic processes (GO:0030244) (Figure 6B). A large number of genes related to cell wall synthesis were enriched, which was consistent with abnormalities in mutant cell walls.

We discovered a substantial number of DEGs that were involved in cell wall synthesis in terms of starch and sucrose degradation and in processes connected to lignin biosynthesis, cellulose biosynthesis, cellulose metabolism, and primary cell walls (Figure 6C). With the exception of *Zm00001eb036680*, all genes associated with secondary cell wall synthesis were specifically expressed in stalks, and the majority were up-regulated in mutant plants (Figure 6D). Additionally, the majority of *ZmBK2* homologous genes expressed in stalks and leaves were cellulose synthase genes, with higher expression levels in stalks (Figure 6E). Genes involved in cell wall development were differentially expressed, suggesting that *ZmBK2* played a critical role in cell wall formation.

### 3.8. Validation of RNA-seq

We selected DEGs involved in cell wall formation for RT-qPCR validation. These included secondary cell wall synthesis genes, glycosyl hydrolase genes, and cellulase genes (Figure 7). Genes for RT-qPCR expression in the RNA-seq are shown in Appendix A. RT-qPCR results also revealed that the expression levels of the selected genes were consistent with those determined by RNA-seq.

## 4. Discussion

### 4.1. Abnormal Cellulose Synthesis Affects Plant Brittleness

Cellulose has a very simple chemical nature, although its assembly may appear complex due to the presence of multi-scale cellulose fibers [32]. Crystalline cellulose elemental fibers composed of glucan chains are bundled into nanofibers, which are then aggregated into cellulosic fibrils. These multiscale cord-like structures constitute a fibrillar network that forms part of the cell wall nanostructure. Therefore, cellulose is a load-bearing polymer of the plant cell wall and can improve the mechanical strength of stalks [33,34]. Many brittle mutants have been identified to date, such as irregular xylem (*irx1*, *irx2*, *irx3*, and *irx5*) in *Arabidopsis* and brittle culm (*bc3*, *bc5*, *bc6*, *bc7*, *bc10*, *bc11*, *bc12*, *bc13*, *bc14*, *bc15*, *bc17*, *bc18*, and *bc88*) in rice [7,10,35,36]. The biochemical and molecular characteristics of these mutants suggest that the brittle phenotype is caused by defects in cellulose synthesis [37]. Cellulose synthase catalytic subunit genes are crucial proteins that directly catalyze glucan chain elongation at the plasma membrane (PM). The COBRA-like protein BC1 regulates cellulose crystallite size and acts as an auxiliary protein to stack nascent glucan chains into a crystalline structure, which is a crucial step in the assembly of cellulose [2,38]. Our RNA-seq results revealed altered CESA expression, demonstrating that catalysis of glucan chain elongation in these subunits is inhibited, which may thus not be normally stacked into a crystalline structure. Therefore, *ZmBK2* may not modulate cellulose assembly by interacting specifically with crystalline cellulose, as the cellulose content was substantially reduced and stalk strength was decreased.

### 4.2. Lignin Synthesis Compensates for Cellulose Deficiency in Mutants

Recent studies have shown that 10 specific enzymes are required for lignin biosynthesis: phenylalanine ammonia-lyase (PAL), cinnamic acid 4-hydroxylase (C4H), 4-coumarate CoA ligase (4CL), cinnamyl alcohol dehydrogenase (CAD), p-hydroxycinnamoyl-CoA (HCT), p-coumarate 3-hydroxylase (C3H), caffeoyl-CoA O-methyltransferase (CCoAOMT), hydroxycinnamoyl-CoA reductase (CCR), ferulate 5-hydroxylase (F5H), and caffeic acid/5-hydroxyferulic acid O-methyltransferase (COMT) [39,40]. Phenylalanine ammonia-lyase is a key enzyme that connects primary and phenylalanine metabolism, catalyzes L-phenylalanine deamination to trans-cinnamic acid, and then converts it to p-coumaric acid through C4H [41]. *Zmpal*, *Zm4cl*, *Zmhct*, *Zmc3h*, *Zmf5h*, and *Zmcomt* are differentially expressed genes that may affect the lignin synthesis process (Figure 8) [42]. Similar to rice *bc7*, *bc10*, *bc11*, and sorghum *bc1* [43], *bk2* decreased cellulose content and slightly increased lignin content in stalks. Through trans-membrane sensor proteins, *SBbc1* senses alterations to the cell wall structure and mechanical properties; in response, these proteins trigger compensatory mechanisms via activating TFs [44]. The compensatory mechanism of rice *bc7* balances the decrease in cellulose content with an increase in hemicellulose, lignin, and silicon content. This dynamic may contribute to the restoration of normal cell wall integrity and maintaining plant growth [45]. Similarly, *bk2* may have a compensatory mechanism in cell wall formation for cellulose deficiency. The synthesis of cellulose and lignin is a dynamically balanced process that involves many key enzymes in metabolism. Defects in any of these key enzymes may affect the abnormal synthesis of cellulose and lignin, ultimately affecting plant brittleness [46,47,48].

### 4.3. ZmBK2 Is a Member of the COBRA Family

Plant *COBRA* genes encode GPI-anchored proteins involved in biological processes related to cell wall biosynthesis and cell expansion [16]. The *AtCOBL4* mutant is characterized by defective cellulose synthesis and decreased cellulose content in secondary cell walls [49]. As putative *AtCOBL4* orthologs, rice *bc1* and maize *bk2* exhibit disrupted secondary cell wall biosynthesis [24]. Rice *bc1* encodes a GPI-anchored protein and disrupts post-translational processing [38]. A 51-bp insertion in *mu-J1862* and a 34-bp deletion in *mu-J184* produced truncated proteins without the GPI attachment site. *ZmBK2* may not properly encode COBRA-like proteins that influence secondary cell wall structure by blocking fibrillin deposition, which decreases cell wall thickness and cellulose content [24]. The *bk2* mutant is derived from a transposon insertion [22], while *mu-J1862* and *mu-J184* are two novel spontaneous mutants that enrich the mutant type of the COBRA gene family and provide richer material for studying cell wall synthesis.

### 4.4. Regulatory Network for Cell Wall Synthesis

Plant cell walls are dynamic and complex networks that require strict coordination among cellulose, lignin, xylan, and other components to execute biosynthetic programs and complete proper cell wall assembly [50]. Specific TFs and other proteins are implicated in coordinating cell wall biosynthesis [51]. In this study, the important DEGs were used to build a regulatory network for cell wall synthesis, and wall-associated kinases (WAK) were at the center of the regulatory network (Figure 9). These protein kinases attach to the cell wall and are important for cell wall plasticity [52,53,54]. However, they are differentially expressed in multiple tissues; WAKs were implicated in cell wall signaling, gene regulation, and pathogen response [55,56]. Three WAK genes (*Zm00001eb139070*, *Zm00001eb071980*, and *Zm00001eb335510*) were down-regulated and may work with *ZmBK2* to regulate cell wall extension and reduce cell wall thickness.

Recent *Arabidopsis* research has contributed to a thorough analysis of TFs involved in cell wall regulation, including subgroup members of the NAC, MYB, and SND [57,58,59]. The NAC TFs function as master switches for secondary cell wall biosynthesis and can activate downstream TFs such as MYB20, MYB42, MYB46, MYB83, MYB103, SND2, SND3, and KNAT7 [60,61]. Some MYB TFs act as second- or third-level master regulators to control the expression of secondary cell wall-related genes [62,63]. For example, MYB46 in *Arabidopsis* is a major switch for secondary cell wall formation and directly regulates the cellulose synthase genes: *ATCESA4*, *ATCESA7*, and *ATCESA8* [64]. MYB46 and MYB83 inhibit the thinning of secondary cell walls by regulating the expression of *SND1* and its homologs *NST1*, *NST2*, *VND6*, and *VND7* [65]. The analysis of DEGs in stalks and leaves by transcriptome showed that 362 TFs were differentially expressed (Appendix A), and the majority belonged to the bHLH, C2H2, ERF, MYB, NAC, and WRKY gene families (Appendix A). Twenty-four TFs, including two NAC genes and four MYB genes were co-down-regulated in stalks and leaves. Among these, NAC35 and NAC110 are likely to be master switches for cell wall biosynthesis to inhibit multiple TFs (MYB56, MYB59, MYB69, and MYB74). Cellulose synthesis-related genes (*CESA1*, *CESA5*, and *CESA9*) were down-regulated. Cellulose synthesis and assembly may be abnormal, which ultimately affects the synthesis of secondary cell walls. Similarly, bHLH152 and bHLH148 are activated as a class of TFs to participate in lignin synthesis.

In the cell wall regulatory network, multiple proteins act synergistically in signal transduction and the regulation of gene expression. The potential interacting proteins of *ZmBK2* were analyzed using an online tool, STRING V11, and eight genes were predicted: *Zm00001eb153340*, *Zm00001eb317090*, *Zm00001eb123640*, *Zm00001eb295810*, *Zm00001eb307070*, *Zm00001eb416400*, *Zm00001eb023530*, and *Zm00001eb421850* (Figure 10). *Zm00001eb153340*, *Zm00001eb295810*, *Zm00001eb416400*, and *Zm00001eb307070* are closely related to cell wall synthesis and encode the catalytic subunits of cellulose synthase, NAC-type TFs, MYB DNA-binding domain superfamily proteins, and WAKs, respectively. These four genes function in the cell wall regulatory network and were differentially expressed in J1862 and *mu-J1862* (Appendix A). ZmBK2 may interact with them to participate in the expression of other genes that are regulated by TFs in the cell wall. Thus, the plant cell wall is regulated by a variety of physiological and biochemical activities, the regulation mechanisms of which still remain unclear.

### 4.5. ZmBK2 Can Potentially Reduce Residue Rate in Fresh Maize

Fresh maize is becoming an increasingly popular foodstuff for its desirable taste and unique qualities, including thin skin, low residue, moderate waxiness, and sweetness [66]. The *bk2* strain has low cellulose content and thus great potential to reduce the residue rate. We measured the residue rate of wild-types and mutants using kernels after 24 days of pollination. The results were as expected: the residue rate of the wild-type was 14.4%, while the mutant had only 12.6% residues. *ZmBK2* can therefore be considered a potentially valuable gene resource for a decrease in residue rate. By controlling *ZmBK2* expression in different parts of the maize plant via genetic engineering, it may be possible to design good-tasting and lodging-resistant maize varieties.

## 5. Conclusions

This study identified a maize mutant with whole-plant brittleness and low cellulose content, leading to reduced lodging resistance. These characteristics were attributable to abnormal cell wall development. Through map cloning and allelic tests, we confirmed that *Zm00001eb393070* encodes a COBRA-like protein and is a target gene for plant brittleness. Transcriptome analysis identified multiple genes related to cell wall development that were used to construct a regulatory network for cell wall synthesis. Our study thus provides a theoretical basis for the mechanisms underlying cell wall development.

## Figures and Tables

**Figure 1 genes-14-01126-f001:**
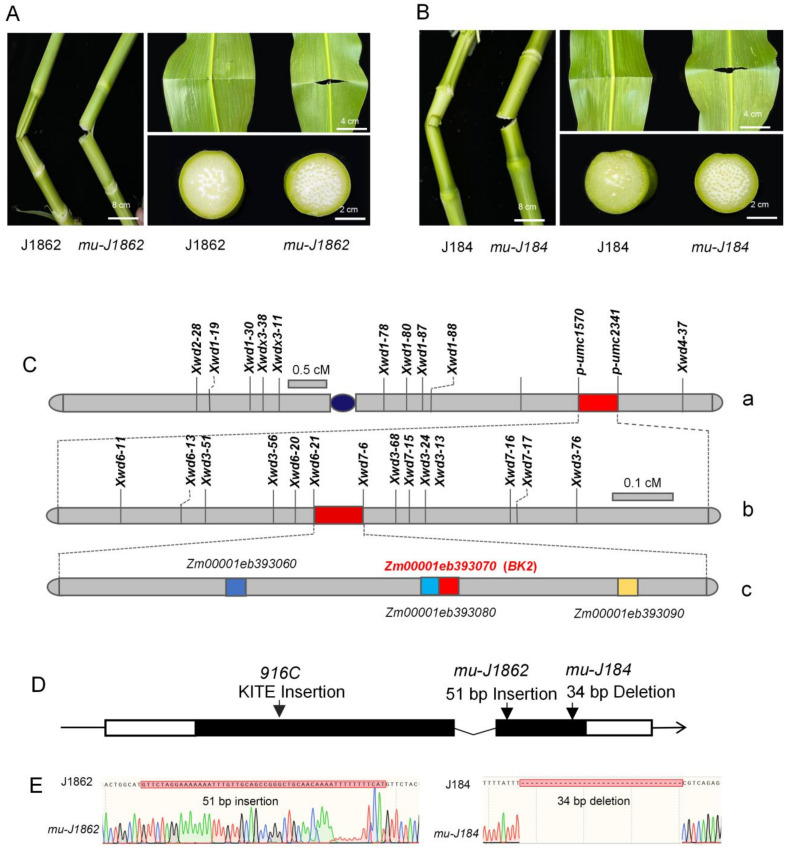
Phenotypes of two maize brittle mutants and map-based cloning of *ZmBK2*. (**A**,**B**) Phenotypes of *mu-J1862* and *mu-J184*. (**C**) *ZmBK2* was mapped to the long arm of chromosome 9 between markers *Xwd6-21* and *Xwd7-6*. Bar indicates the genetic distance in cM. The localization interval contains four OFRs, and the target gene *ZmBK2* is highlighted in red. (**D**) *ZmBK2* structure and mutation sites of *mu-J1862*, *mu-J184*, and *916C*. The white boxes represent the 5′ and 3′ UTR, the black boxes represent coding sequences, and the line graph line between boxes represents introns. The arrows indicate the mutation sites of *mu-J1862*, *mu-J184*, and *916C*. (**E**) The sequencing peak map of *mu-J1862* and *mu-J184*.

**Figure 2 genes-14-01126-f002:**
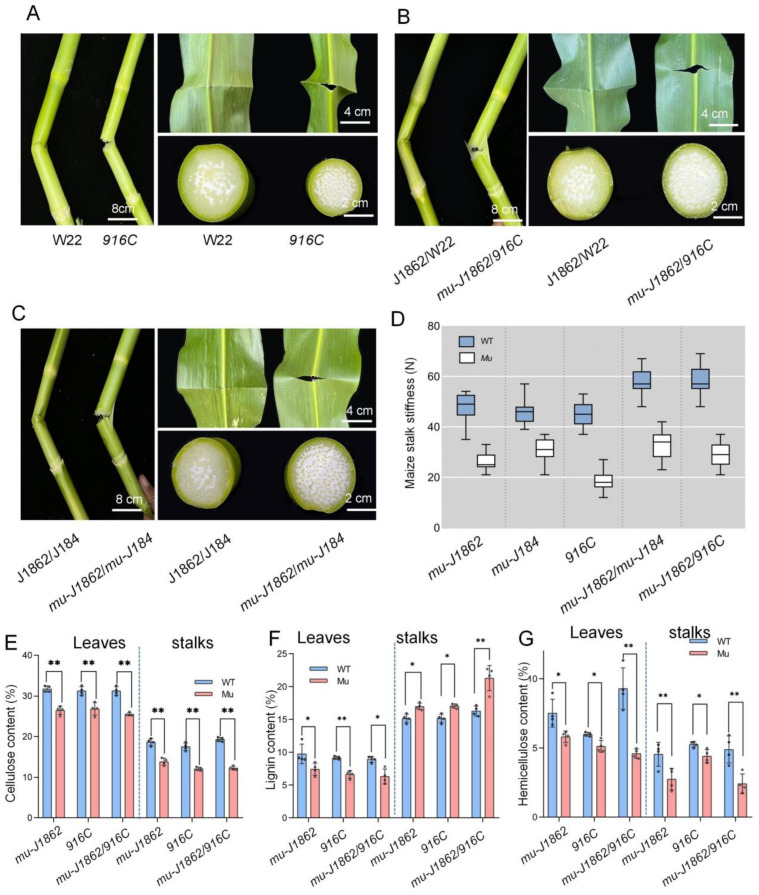
Allelic tests and composition measurements of stalks and leaves. (**A**–**C**) Phenotypes of *916C*, *mu-J1862*/916C, and *mu-J1862*/*mu-J184*. (**D**) Stiffness testing of the third internode above the ground for *mu-J1862*, *mu-J184*, *916C*, *mu-J1862*/*mu-J184*, and *mu-J1862*/*916C*. (**E**–**G**) Cellulose, lignin, and hemicellulose determination in leaves and stalks of *mu-J1862*, *916C* and *mu-J1862*/*916C* after appearance of the brittle phenotype. The values are presented as means ± SD and were compared using Student’s *t*-test (* *p* < 0.05, ** *p* < 0.01). WT: wild-type, Mu: mutants.

**Figure 3 genes-14-01126-f003:**
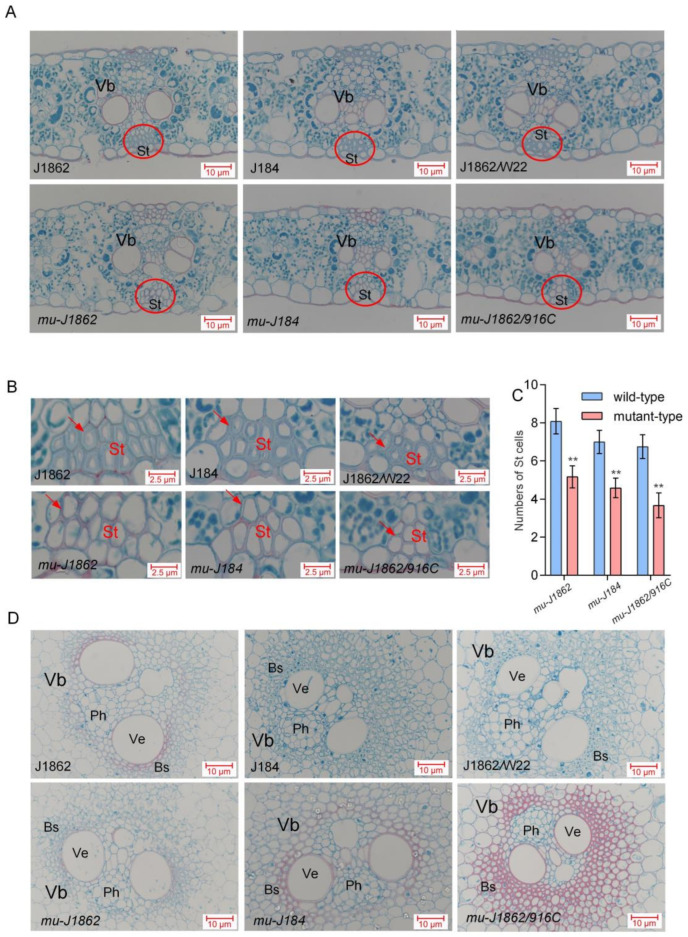
Paraffin section observation of leaves and stalks (**A**) Paraffin sections of *mu-J1862*, *mu-J184*, and *mu-J1862*/*mu-916C* leaves. (**B**) Enlarged view of sclerenchyma cells marked by circles in (**A**). (**C**) Cell number count of sclerenchyma cells. The values are presented as means ± SD and were compared using Student’s *t*-test (** *p* < 0.01). (**D**) Paraffin sections of *mu-J1862*, *mu-J184*, and *mu-J1862*/*mu-916C* stalks. Vb: vascular bundle, St: sclerenchyma tissue, Ph: Phloem, Ve: vessel, Bs: bundle sheath, WT: wild-type, Mu: Mutants. The red arrows in (**B**) indicate example sclerenchyma cells.

**Figure 4 genes-14-01126-f004:**
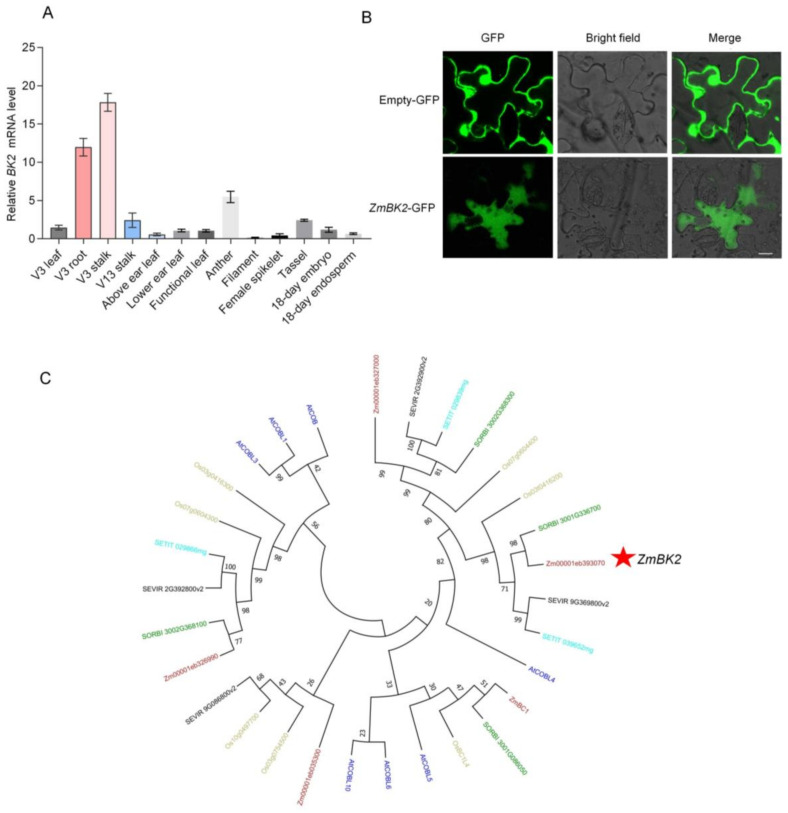
Expression patterns of *ZmBK2*. (**A**) Expression of *ZmBK2* in various types of tissues. The results are means ± SD of triplicate assays. (**B**) *ZmBK2* was fused with eGFP, and the subcellular localization of *ZmBK2* was analyzed by its transient expression in tobacco leaves. The bright green coloration indicates that the *ZmBK2* protein is primarily localized in the cytoplasm (**C**) The maximum-likelihood tree based on 44 *ZmBK2* homologs protein sequences is rooted in *AtCOBL4*. The amino acid sequences were obtained from a NCBI database search, and aligned using MEGAX. Scale bar represents 0.2 substitutions per site. Red, *Zea mays*; dark blue, *Arabidopsis*; black, *Setaria viridis*; sky blue, *Setaria italica*; green, *Sorghum bicolor*; yellow, *Oryza sativa*.

**Figure 5 genes-14-01126-f005:**
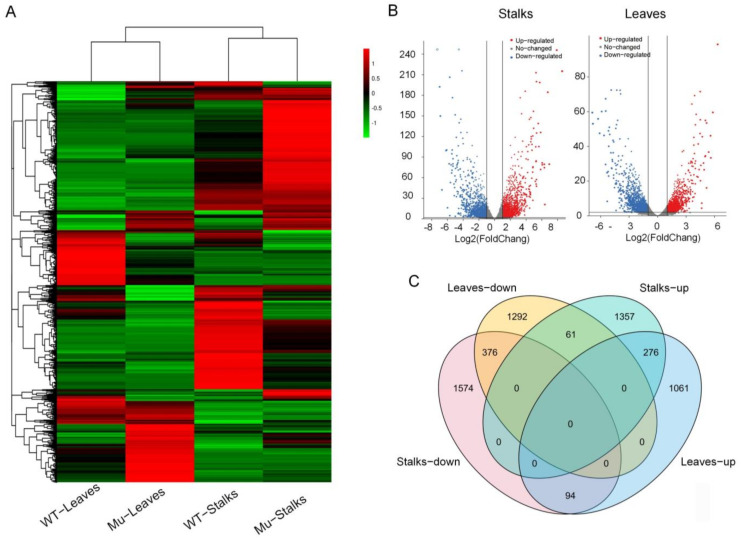
Overview of differentially expressed genes (DEGs) between mutants and wild-types. (**A**) Heatmap of 6942 DEGs between mutants and wild-types. The expression values were normalized by setting the mean of every row to zero and the standard deviation of every row to one. (**B**) Volcanic plot of differentially expressed genes in stalks and leaves. (**C**) Venn diagram of total DEGs (Padj < 0.01) between genotypes. WT: wild-type, Mu: mutants.

**Figure 6 genes-14-01126-f006:**
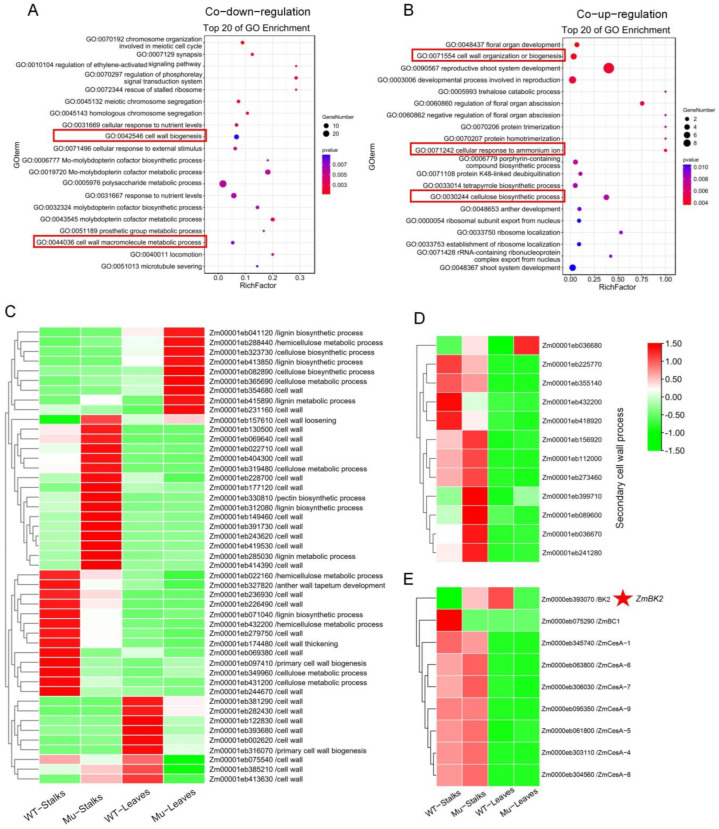
Differentially expressed genes analysis (DEGs). (**A**) Gene oncology (GO) enrichment analysis of co-down-regulation genes in leaves and stalks. (**B**) GO enrichment analysis of co-up-regulation genes in leaves and stalks. (**C**–**E**) Heatmap of differential expression genes of the cell wall, secondary cell wall processes, and the homologous gene of *ZmBK2*. WT: wild-type, Mu: mutants.

**Figure 7 genes-14-01126-f007:**
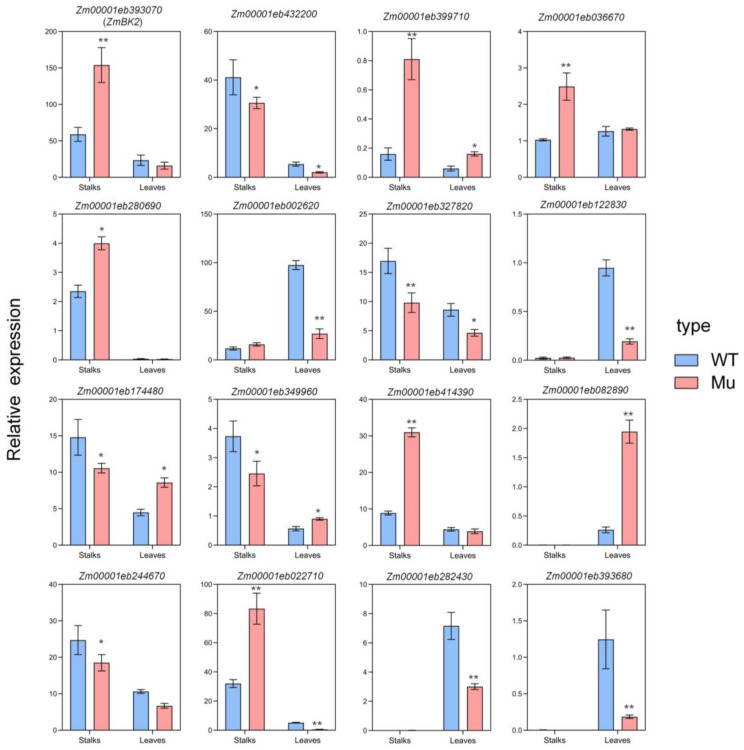
Differentially expressed genes (DEGs) involved in cell wall metabolism were chosen for RT-qPCR validation. The relative expression of *ZmBK2* genes in various tissues of mutant and wild-type plants is shown. The values are presented as means ± SD and were compared using Student’s *t*-test (* *p* < 0.05, ** *p* < 0.01). Mu (pink): mutants; WT (blue): wild-type.

**Figure 8 genes-14-01126-f008:**
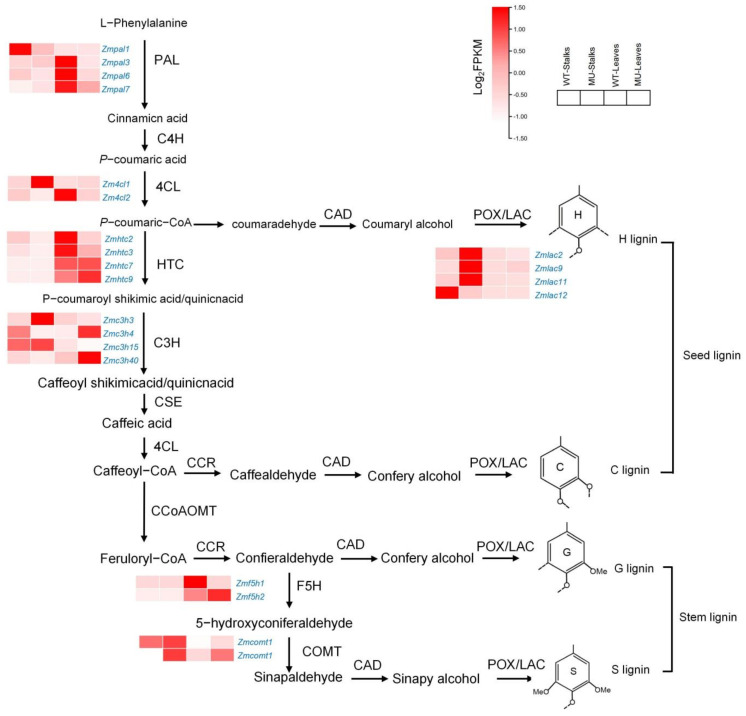
Key differentially expressed genes (DEGs) in the lignin metabolic network.

**Figure 9 genes-14-01126-f009:**
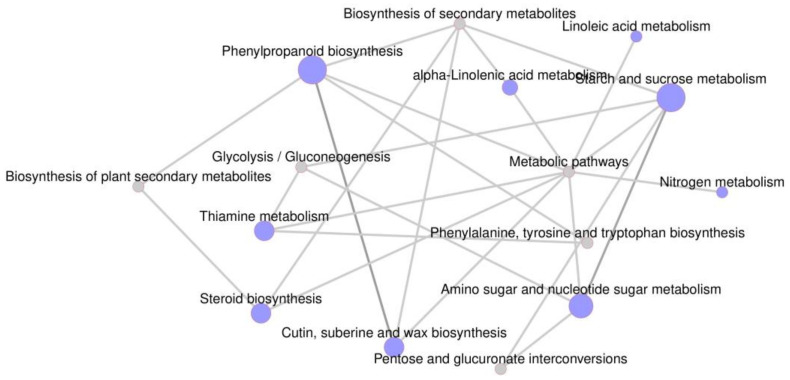
Network graph showing GO terms in biological processes.

**Figure 10 genes-14-01126-f010:**
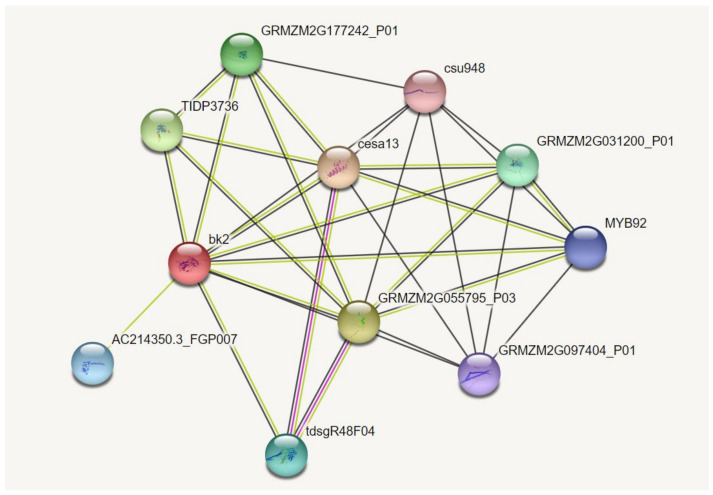
Interaction network graph between *ZMBK2* gene and other factors/genes, as predicted by STRING.

**Table 1 genes-14-01126-t001:** Genetic analysis of the brittle gene.

Group	Total Numbers	Wild-Type	Mutant Type	χ^2^(0.05) ≤ 3.84
(B73/*mu-J1862*) F_2_	672	511	161	0.389
(Huang Zaosi/*mu-J1862*) F_2_	457	338	119	0.26

## Data Availability

Data are openly available in a public repository. The raw data of transcriptome sequencing in this study are provided at the NCBI short read archive (accession number: PRJNA896366).

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
