# Peer review of "Identification of ZmBK2 Gene Variation Involved in Regulating Maize Brittleness"

_genes, 2023, doi:10.3390/genes14061126_

Round 1

Reviewer 1 Report

I find this study interesting and the manuscript very well written. The authors have identified a maize mutant gene ZmBK2 that is associated with a defective cellulose synthesis pathway and thus has reduced plant mechanical strength. This finding adds to the understanding of the maize cell wall regulatory network and, together with other identified mutants, provides the possibility of future studies to increase lodging resistance in maize. Overall, the manuscript is strong: the background knowledge provided in the Introduction section is sufficient; the experimental designs and methods are clearly presented; the results and conclusions are sound and convincing.

Reviewer 2 Report

The study was focused on identification of ZmBK2 gene variation involved in regulating maize brittleness. The Authors stated that map-based cloning and allelic tests identified a maize mutant associated with decreased stalk strength, and confirmed that the mutated gene, ZmBK2, is a homolog of Arabidopsis AtCOBL4, which encodes a COBRA-like glycosylphosphatidylinositol (GPI)-anchored protein. The bk2 exhibited lower cellulose content and whole-plant brittleness. Microscopic observations showed that sclerenchymatous cells were reduced in number and had thinner cell walls, suggesting that ZmBK2 affects the development of cell walls.

In my opinion, the paper is quite interesting in the research field. However, I recommend the following improvements:

-        The Introduction is overloaded in the content,

-        I strongly suggest including the electropherograms presenting the RNA bands in agarose gels in the manuscript or in the Supplementary file – it would provide information regarding quality of RNA samples,

-        In addition, RIN numbers of RNA samples should be presented in the manuscript (the RNA Integrity Number = RIN),

-        The Authors used SYBR Green fluorescent dye during gene expression studies, hence, it is obligatory to perform Melting Curve Analysis, and results of this examination should be added in the manuscript or Supplementary file (e.g., JPG or TIFF file).

-        Student's t-test is inappropriate for testing the set of data in the manuscript. A factorial ANOVA with a follow-up post-hoc test (e.g. Tukey's) should be used,

-        Fig.7 – results of statistical analyses should be presented in the graph,

-        Some newer citations in the research field should be included, and the older ones may be discarded,

-        Moderate English changes by the native speaker specialist are required.

Moderate English changes by the native speaker specialist are required.

Reviewer 3 Report

Dear authors, 

The manuscript genes-2404401 titled ‘Identification of ZmBK2 gene variation involved in regulating maize brittleness’ was interesting. Authors identified genes concerned with cell wall development and provided a new insight into the mechanisms of maize brittleness. I am providing some minor comments to this manuscript, and hope it will useful for improving it.

Minor comments

1.     In the section 2.1, could authors explain little about the B73, Huangzaosi, and W22. Why did authors use these maize lines to conduct cross experiment?

2.     In line 159, please add the full name of V3.

3.     Please try to make the Figure 1E and 1F clearer, they are fuzzy.

4.     I noticed that mu-J1862, mu-J184, 916C, mu-J1862/916C, mu-J1862/mu-J184 have more white points in the cross section of leave’s vein, are there some relationships between the white point with the brittle of maize?

5.     I suggest authors to explain Figure5A and Figure5B in the result section, because you have mentioned the heatmap and volcanic plot of differently expressed genes.

Dear authors, please check the English again by native-English speaker.
